# Exposure to Tobacco Smoking in Vehicles, Indoor, and Outdoor Settings in Germany: Prevalence and Associated Factors

**DOI:** 10.3390/ijerph19074051

**Published:** 2022-03-29

**Authors:** Martin Mlinarić, Sabrina Kastaun, Daniel Kotz

**Affiliations:** 1Medical Faculty, Institute of Medical Sociology (IMS), Martin Luther University Halle-Wittenberg, 06112 Halle (Saale), Germany; martin.mlinaric@gmail.com; 2Institute of Applied Marketing and Communication Studies (IMK), 99084 Erfurt, Germany; 3Institute of General Practice, Addiction Research and Clinical Epidemiology Unit, Centre for Health and Society, Medical Faculty of the Heinrich-Heine-University Düsseldorf, 40225 Düsseldorf, Germany; sabrina.kastaun@med.uni-duesseldorf.de

**Keywords:** secondhand smoke exposure, vehicles, indoor, outdoor, social epidemiology, household survey

## Abstract

Little is known on whether secondhand smoke (SHS) exposure in vehicles, indoor, and outdoor settings is similarly patterned in terms of different socio-epidemiological indicators in Germany. This study aims to estimate the current national-level prevalence and associated socio-epidemiological indicators of SHS exposure in vehicles, indoor, and outdoor settings in the German population, using current data from a representative household survey. We used cross-sectional data (N = 3928 respondents aged 14–99 years) from two waves of the DEBRA survey (German Study on Tobacco Use), conducted between January and March 2020. The reported prevalence of SHS exposure during the last seven days was 19% in vehicles, 25% in indoor settings, and 43% in outdoor settings. We found that younger age and current smoking were consistently associated with higher SHS exposure. Furthermore, people with low education were more likely to be exposed to SHS in vehicles and indoor settings than people with high education. This study found that the prevalence of SHS exposure in vehicles, indoor, and outdoor settings is a relevant feature of everyday life in Germany, especially for younger people and people with lower education, leading to potentially persistent socioeconomic and tobacco-attributable inequalities in morbidity and mortality.

## 1. Introduction

Secondhand smoke (SHS) exposure in small, enclosed spaces has serious health effects (e.g., decreased lung function, asthma, persistent wheezing, sudden infant death) [1,2]. Currently, about 20% of adolescents [3,4,5] and adults [6] across high-income countries are exposed to SHS exposure in small, enclosed spaces such as cars or at their homes [7]. SHS exposure in indoor settings (e.g., in the private vehicle) is comparable to airborne concentration in indoor smoking bars, although atmospheric and biological markers of SHS concentration may be—in the case of private vehicles, for instance—mediated by the air conditioning, extent of airflow, and number of inches the windows are open [1,2,8,9].

Exposure to SHS in vehicles is correlated with future smoke initiation (primary prevention relevance) among youth and adolescents [1]. Exposure to SHS at public, outdoor places (e.g., at parks, bus stops, playgrounds, sport and recreational facilities) may be less harmful and toxic with regard to adverse health outcomes, but exposure to, and visibility of, smoking is linked to the societal degree of tobacco “de-normalisation” [10] and the establishment of public role models for children and adolescents [11], as well as to positive beliefs about and public acceptance of smoking [12,13]. Public acceptance of smoke-free regulations varies substantially across settings [14,15], with regulations for non-smoking indoors and in the presence of minors (“child frame” [16]) reaching the highest support rates, while regulations for public, outdoor or adult settings (e.g., parks, outdoor workplaces, public outdoor events) are more contested and receive less support [13,14,17].

The introduction of smoke-free car legislation in Germany is supported by a vast majority of the general population (72%), and even by 67% of current smokers [18], which indicates that the implementation of such a policy, which already exists in other European countries—such as Ireland, France, Italy, or the United Kingdom—might be feasible in Germany [15,18]. However, even if public support rises for indoor or children-related areas, such as schools, bars, or public buildings, in Germany, only about 38% of the population support smoking bans in public parks or recreational facilities [15]. Therefore, outdoor settings probably remain the final “frontier” [13] of smoke-free policy innovation, also in Germany.

For at least 15 years we have observed declining smoking prevalence and SHS exposure at the workplace [15] and at home in Germany [19,20,21]; this holds also for indoor settings and cars in similarly high-income countries such as the USA [22,23]. However, approximately 26% of adults in the EU [24] and 28% of the 14-years-and-older population in Germany [25] are still current smokers.

Smoking inequalities by sex, race, and socioeconomic status (SES) are widely documented as well [3,5,20,21,22]. There are persistent inequalities by SES in SHS exposure and successful quit attempts [21,26], corresponding with SES inequalities in exposure to car smoking in youth [5,27]. Inequalities hinder smoke-free policy implementation at home or in private vehicles, as people with low SES are more likely to be smokers, and successful implementation in the private, family setting often depends on the parental SES/smoking status [28].

However, there are wide geographic differences and ranges in SHS exposure in vehicles even across European cities [5] that are located in national policy environments where smoking in vehicles is banned while children or pregnant women are present (e.g., SHS exposure in cars on at least one day in the last week: Italy: 44% vs. Finland: 7%) [5]. In Italy, the prevalence of smoking while driving is 66% among adult smokers [29]. Even in advanced-tobacco-control countries, such as Canada, one can observe substantial differences between places (e.g., British Colombia: 16% vs. Saskatchewan: 37%) [30]; therefore, prevalence rates are dependent on the national as well as regional/local context.

So far, research on SHS exposure has focussed on youth or adolescents and Anglophone countries [1,2,3,4,30,31]. Moreover, SHS exposure is often exclusively studied either in the indoor setting, at outdoor places, or in cars, but not all together in one study. Little is known on whether exposure to smoking in indoor (e.g., SHS exposure in vehicles) and outdoor settings is similarly patterned in terms of different socio-epidemiological indicators (e.g., SES, age, migration) in Germany. In Germany, slow progress has been made in tobacco control during the past two decades, and the tobacco-prevention environment is relatively defensive compared to similar, high-income countries due to partial and inconsistent smoke-free policies in the hospitality sector and barriers in the multi-level governance of the federal system, low prices for tobacco products, and no smoke-free car legislation [32]. However, there are things moving as, in May 2021, civil society actors and tobacco-control advocates launched an appeal for a smoke-free Germany by 2040; although, due to the coronavirus pandemic, prioritisation of tobacco prevention and regulation by the federal health ministry faces additional difficulties.

### Study Aims

Next to documented public support for smoke-free car policies in Germany [18], we know that convincing evidence on the magnitude of harmful SHS is known to be a precondition for the adoption and sustained implementation of smoke-free policies at indoor, private, and outdoor settings [33]. Therefore, we aim to contribute updated and new evidence on the German situation, which may stimulate and strengthen smoke-free car policy and comprehensive, smoke-free indoor (e.g., hospitality sector) or outdoor (events frequented with minors) ambitions to prevent SHS and smoking normalisation. We used recent data from the ongoing German Study on Tobacco Use (DEBRA) [33]—a nationally representative household survey to:estimate the current national-level prevalence of SHS exposure in vehicles, indoor, and outdoor settings in the German population;estimate the current prevalence of SHS exposure in the above-named settings in relation to socio-epidemiological indicators of inequality (age, sex, level of education, household income, migration background) and tobacco-smoking status; and toexplore independent associations of socio-epidemiological indicators of inequality and tobacco-smoking status with SHS exposure in the above-named settings.

## 2. Materials and Methods

The DEBRA study (DEBRA. Available online: http://www.debra-study.info, accessed on 22 March 2022)—a representative, Germany-wide, computer-assisted, face-to-face household survey of individuals aged 14 years and older—was initiated in June 2016 (see study protocol [34]) and collected data on key indicators, such as current tobacco and alternative nicotine product (e.g., e-cigarettes) use, attempts to quit smoking, and the use of methods to support smoking cessation. Respondents were selected by using a dual-frame design: a composition of random stratified sampling (50% of the sample) and quota sampling (50% of the sample). The sampling design is described in detail elsewhere (DEBRA quota sampling. Available online: https://osf.io/s2wxc/, accessed on 22 March 2022). The DEBRA study was reviewed by the ethics committee of the Heinrich Heine University, Düsseldorf (ID 5386/R), and registered in the German Registry of Clinical Trials (DRKS00011322, DRKS00017157).

### 2.1. Sample

This article presents the aggregated data (N = 3928) from waves 22 and 23 of the DEBRA survey, conducted in January 2020 (n = 2061) and February/March 2020 (n = 1867). The rationale for using these two waves was grounded in the reason that SHS exposure in vehicles was queried only in wave 22, whereas exposure at indoor and outdoor places was queried as part of the following wave 23.

### 2.2. Outcome Measures

The prevalence of SHS exposure in vehicles was measured by asking respondents “On how many of the past 7 days have you been in a vehicle, privately or professionally, while someone was smoking tobacco in it, i.e., cigarette or cigar or pipe? Not meant are e-cigarettes or other vaporisers. Please estimate the number of days.” With answer options (1) “on no day at all”, (2) “on 1 or 2 days”, (3) “on 3 or 4 days”, (4) “on 5 or 6 days”, (5) “on all 7 days”, (6) “I have not driven in any vehicle in the last 7 days”, (7) “don’t know”, and (8) missing answer. Respondents with answer options 6 to 8 (n = 296 of 2061) were excluded from the analyses.

For SHS exposure in indoor settings, people were asked “On how many of the past 7 days have you been in closed indoor rooms and inhaled the smoke of someone who smoked tobacco, i.e., cigarette or cigar or pipe? Not meant are e-cigarettes or other vaporisers. Think for example of your workplace, school or training centre, at your home or at other peoples’ home, or of rooms in public buildings. Please estimate the number of days.” With answer options (1) “on no day at all”, (2) “on 1 or 2 days”, (3) “on 3 or 4 days”, (4) “on 5 or 6 days”, (5) “on all 7 days”, (6) “don’t know”, and (7) missing answer. Respondents with answer options 6 or 7 (n = 79 of 1867) were excluded from the analyses.

Finally, SHS exposure in outdoor settings was operationalised with the question “On how many of the past 7 days have you been outdoors and inhaled the smoke from someone who smoked tobacco, i.e., cigarette or cigar or pipe? Not meant are e-cigarettes or other vaporisers. With outdoors we mean for example bus or train stations, outdoor areas of cafés, bars or restaurants, outdoor areas of your workplace, parks, playgrounds or other places or paths. Please estimate the number of days.” Answer options were the same as described above, and respondents with answer options 6 or 7 (n = 106 of 1867) were excluded from the analyses.

It is possible that the same respondent reported SHS exposure in more than one setting. Hence, our outcomes measured the likelihood of SHS exposure in each of the different settings, rather than the total level or burden of SHS exposure from all settings together.

### 2.3. Socio-Epidemiological Covariates: Sociodemographic Data and Smoking Status

Age in years was used as a continuous variable for regression analyses and as a categorical variable in age groups (14–17, 18–24, 25–39, 40–64, 65+ years) for descriptive statistics. Moreover, sex (male/female) was included as a sociodemographic factor. Furthermore, migration background (yes/no), whether the respondent had at least one parent born in a country other than Germany, was added to the regression models. The question on migration background was optional, and roughly 7% of respondents declined to answer this question. The level of education (low, middle, and high) and the net household income as a continuous variable in EUR per month among over-18-year-olds (EUR 0 to EUR 7000 or more) and as (1) low, (2) middle, (3) high for descriptive statistics, were included as socioeconomic factors for regression analyses and as categorical variables. As the needs and expenses of a household depend on the age and number of people living in it, we used an equalisation technique of the Organisation for Economic Co-operation and Development (OECD) to adjust income for household size and composition (OECD-modified equivalence scale). Each member of a household received a different weighting, and the net total household income was divided by the sum of the weightings to calculate a representative household income. Details on the calculation are published online (Calculation OECD-modified household income. Available online: https://osf.io/387fg/, accessed on 22 March 2022). Lastly, tobacco-smoking status was added by distinguishing “current”, “ex-smoker”, and “never” smokers of tobacco products (excluding e-cigarettes and heated tobacco products).

### 2.4. Statistical Analyses

Both an analyses plan (DEBRA study protocol passive smoke exposure. Available online: https://osf.io/5xtbk/, accessed on 22 March 2022) and the statistical code (DEBRA SHS exposure v3.sps. Available online: https://osf.io/ra97e/, accessed on 22 March 2022) was written and published prior to the statistical analyses. The data were weighted to be representative of the German population, accounting for personal and household characteristics, and the weighted data were used for the descriptive analyses of the main outcomes (Table 1, Table 2 and Table 3). Details on the weighting technique are described in the study protocol [34]. The regression analyses displayed in Table 4 used unweighted data. The associations between SHS exposure and socio-epidemiological indicators were analysed using three separate multivariable linear regression models presenting regression coefficients with 95% confidence intervals (CIs). The statistical analyses followed three steps:

First, descriptive analyses of weighted prevalence rates with 95% CIs of SHS exposure in vehicles, enclosed, indoor settings, and at public, outdoor places were performed. Second, descriptive analyses of weighted prevalence rates with 95% CIs of SHS exposure in the above-named settings by age (as categorical variable), sex, level of education, household income (as categorical variable), migration background, and tobacco-smoking status were calculated. Third, three separate multivariable linear regression models for the respective outcome measures (SHS exposure per setting: from “on no day” to “on all 7 days”), each with independent covariates, age (as continuous variable), sex, level of education, household income (as continuous variable), migration background, and tobacco-smoking status were analysed.

Missing data on independent variables were assumed to be low due to the face-to-face method of data collection and turned out to be very low (<1%) per variable. We applied complete case analyses only, which led to slightly different sample sizes per regression analysis of the respective outcome. Statistical analyses were performed with SPSS (IBM) and R Studio for computing the 95% CIs.

## 3. Results

Table 1, Table 2 and Table 3 display that the reported prevalence of SHS exposure during the last seven days was 19% in vehicles, 25% in indoor settings, and 43% in outdoor settings. A small minority (4 to 7%) reported being exposed to SHS on all seven days of the last week in vehicles and indoor rooms (Table 1 and Table 2), while at least two out of ten smokers reported daily passive tobacco exposure in indoor (Table 2) and outdoor areas (Table 3).

People in the age group of 25 to 39 years were most frequently exposed to SHS on all seven days of the past week in vehicles (7%), indoor (10%), and outdoor (12%) settings. In current smokers, exposure rates on all seven days in the three settings were 9%, 19%, and 22%, respectively.

Associations between socio-epidemiological factors and smoking status and the outcome SHS exposure in the three different settings are reported in Table 4. Across all settings, younger age and current smoking status were consistently associated with higher SHS exposure. According to the descriptive analyses (Table 1, Table 2 and Table 3) and regression models (Table 4), 18- to 24-year-olds and current smokers were the central groups here. No significant differences across migration background can be reported from the regression analyses. Differences by sex were found only for SHS exposure in vehicles (Table 4). Female respondents were less likely to have experienced passive smoking in the past seven days in a vehicle. Educational inequalities were observed with regard to SHS exposure in vehicles and indoor places: persons with lower education were at higher risk of being exposed to SHS compared with people with higher education. However, in outdoor settings, no differences by SES were evident as only smoking status—similarly to vehicles and indoor settings—was a significant correlate for SHS exposure. In comparison to school-leaving qualification, household income did not follow the same SES pattern in vehicles and indoor contexts as, in vehicles, for instance, a positive linear association was found; the higher the income, the more likely the respondent was to have experienced SHS exposure in vehicles in the past seven days (Table 4).

## 4. Discussion

This study investigated whether exposure to smoking in indoor (including exposure in vehicles) and outdoor settings is similarly patterned in terms of different socio-epidemiological indicators. We found, in line with existing studies [5,7,11,29], that younger age and smoking status were consistently associated with higher SHS exposure in vehicles and in indoor and outdoor places, while educational inequalities by highest school-leaving qualification were only relevant for SHS exposure in indoor settings such as vehicles or indoor rooms.

### 4.1. Interpretation of Central Findings

The identified, weighted prevalence of SHS exposure in vehicles on at least one day in the past seven days (19%) was located in the known wide range from existing studies on other European or Anglo-Saxon (e.g., British Colombia: 16% vs. Saskatchewan: 37%) [30] contexts. The SILNE-R study on seven European cities found, for instance, that SHS exposure in cars on at least one day in the last week was reported, on average, by 22% of 14- to 17-year-old adolescents, while, in the German city of Hanover, 19% of this age group reported an identical prevalence [5], comparable to our study on the general German population. Additionally, across other high-income countries, approximately one fifth of youths [3,4,5,35] and adults [6] report SHS exposure in cars or at home [7].

We found that 11% of the respondents, and even 9% of non-smokers, were exposed to SHS at indoor places on one or two days in the past week (Table 2). According to data provided by the Robert Koch Institute (RKI) from a German, representative health survey (GEDA 2014/2015-EHIS), 11% of non-smoking adults were also regularly exposed to passive smoking in enclosed spaces, and this was particularly the case with young adults, which corresponds with our findings on age in the indoor setting. Non-smoking females seemed to be more often exposed to SHS when socialising with friends and acquaintances (51%), whereas non-smoking males faced passive smoking in the workplace (56%) [36].

The relatively high visibility and prevalence of SHS exposure outdoors (43%) identified in this study corresponds with the evidence that smoke-free regulations receive far less support for public, outdoor or adult settings (e.g., parks, outdoor workplaces, or public, outdoor events) [13,14,17]. Therefore, de-normalisation of tobacco smoking is substantially different between indoor and outdoor spheres, and places, such as parks or recreational areas, remain the final “frontier” [13] of smoke-free policy innovation. However, a recent European study from the EUREST-PLUS ITC Europe Surveys suggested that SHS exposure in public places is significantly less likely in countries with total bans as compared to those countries with partial bans [37].

As in our study on SHS exposure in three socio-spatial contexts, level of education and age were also reported as major correlates of smoking prevalence in a comparative study on 12 European countries [24]. The social gradient in SHS exposure across both indoor settings is not a surprising result, as socioeconomic inequalities in smoking by education in youth [38,39] and adulthood are a widely documented phenomenon in the literature on tobacco control and smoking prevention [26,36].

### 4.2. Limitations and Strengths

The DEBRA study had methodological limitations, as is to be expected from large national surveys; for example, all data were self-reported. The study design was cross-sectional, and, as such, the relationships observed should be interpreted with caution as no causal interpretation can be inferred from the analyses. In addition, selection and response bias or social desirability may have occurred during the face-to-face data collection. In terms of the measured SHS exposure during the past seven days in the three studied settings, it is difficult estimate or judge on the intensity of the toxic exposure in private vehicles, indoor rooms, and outdoor places. As the data suggest, SHS exposure is not equally distributed across the past seven days and takes mostly place on three or four days, which may influence effects on health outcomes. Moreover, we analysed a variety of sociodemographic and socioeconomic covariates, but there may be even more relevant sociological or psychological factors that influence or mediate higher or lower SHS exposure among certain groups. Therefore, unobserved heterogeneity of the data and DEBRA sample may be an issue of concern. On the other hand, the refined methods of sampling and data weighting permitted an accurate analysis of current data on SHS exposure in three different socio-spatial settings (vehicles, indoors, and outdoors) that is representative of and unique to the German population. Moreover, due to it being a face-to-face survey, only scant data were missing.

Our fieldwork was conducted in the period between January and March 2020 and did not cover the entire year. The level of SHS exposure may be different in different months of a year, e.g., due to the weather or holidays. We do not expect this to have an influence on the associations between SHS exposure and the socio-epidemiological factors under study. However, our estimated prevalence rates of SHS exposure may be different from the true annual average.

A draft pertaining to cars carrying minors and pregnant women was proposed in 2016 by the Federal Drug Office in Germany [40] and was introduced into legislative procedures by the Federal Council in 2019. However, the final adoption by German Federal Parliament still pends, and there is a lack of socio-epidemiological data, especially on SHS exposure in cars and at outdoor places. Tobacco-control efforts are blocked by the anti-tobacco-control lobby, which argues, based on data of the RKI [20,21], that SHS exposure at home and youth smoking has declined. A lot of evidence comes from specific settings and populations (e.g., adolescents/school-aged children) and local levels (e.g., cities/regions/municipalities) [5]. Little is known on the German national level, and this study adds recent data based on the representative DEBRA survey.

## 5. Conclusions

This study found that the prevalence of SHS exposure in vehicles and in indoor and outdoor settings remains a relatively normal and visible feature for younger people and disadvantaged groups in Germany, leading to potentially persistent socioeconomic and tobacco-attributable inequalities in morbidity and mortality. Toxic, passive exposure to tobacco smoke indoors and high visibility of smoking hinder de-normalisation efforts and public health agendas. Therefore, progressive implementation in Germany of the measures drawn up in the WHO Framework Convention on Tobacco Control, such as smoke-free car legislation, advertising bans, comprehensive smoke-free policies in the hospitality sector, and increased tax policies, should be given priority in terms of health policy.

## Figures and Tables

**Table 1 ijerph-19-04051-t001:** Descriptive sample characteristics of SHS exposure in vehicles among people who drove or rode in vehicles.

	Weighted Prevalence of SHS in Vehicles (95% Confidence Interval)				
Days (d) Exposed	0 d	1|2 d	3|4 d	5|6 d	all 7 d
Total					
n = 2061	80.8 (78.9–82.6)	8.7 (7.5–10.2)	3.0 (2.3–3.9)	3.7 (2.9–4.7)	3.7 (2.9–4.7)
Sex					
Male	76.7 (73.8–79.4)	9.1 (7.3–11.2)	4.1 (2.9–5.6)	4.8 (3.5–6.5)	5.3 (3.9–7.0)
Female	85.0 (82.5–87.3)	8.4 (6.6–10.4)	1.9 (1.1–3.1)	2.5 (1.6–3.8)	2.2 (1.3–3.4)
Age					
14 to 17	93.7 (84.5–98.2)	1.6 (0.0–8.5)	3.2 (0.4–11.0)	1.6 (0.0–8.5)	0.0 (0.0–5.7)
18 to 24	71.9 (64.4–78.5)	18.0 (12.5–24.6)	6.0 (2.9–10.7)	1.2 (0.1–4.3)	3.0 (1.0–6.8)
25 to 39	73.2 (68.5–77.5)	11.6 (8.7–15.2)	2.5 (1.2–4.6)	5.8 (3.7–8.6)	6.8 (4.6–9.8)
40 to 64	78.6 (74.5–81.5)	8.8 (6.9–11.1)	3.7 (2.4–5.3)	4.9 (3.4–6.7)	4.1 (2.8–5.8)
65+ years	94.2 (94.0–.98.0)	3.0 (1.6–5.3)	0.8 (0.0–2.2)	1.0 (0.0–2.6)	1.0 (0.0–2.6)
Migration background					
None	84.2 (82.1–86.0)	6.7 (5.5–8.1)	2.5 (1.8–3.5)	2.9 (2.1–4.0)	3.7 (2.7–4.8)
Yes ^1^	72.4 (66.9–77.4)	15.7 (11.7–20.4)	3.1 (1.4–5.8)	6.1 (3.7–9.5)	2.7 (1.2–5.3)
Education					
High	82.9 (79.6–85.8)	8.5 (6.4–11.0)	2.9 (1.7–4.6)	2.5 (1.4–4.2)	3.2 (1.9–5.0)
Middle	77.1 (73.5–80.4)	9.5 (7.3–12.1)	3.8 (2.4–5.6)	5.2 (3.6–7.3)	4.4 (2.9–6.4)
Low	80.4 (76.5–83.9)	8.7 (6.3–11.6)	2.8 (1.5–4.7)	3.8 (2.3–6.0)	4.3 (2.6–6.5)
Income					
High	81.6 (77.3–85.3)	7.5 (5.1–10.6)	1.8 (0.7–3.7)	3.9 (2.2–6.3)	5.2 (3.2–7.9)
Middle	81.5 (79.1–83.6)	8.3 (6.8–10.0)	3.1 (2.2–4.3)	3.7 (2.7–4.9)	3.5 (2.9–4.7)
Low	74.6 (67.8–80.6)	14.3 (9.6–20.1)	4.8 (2.2–8.8)	3.7 (1.5–7.5)	2.6 (0.9–6.1)
Smoking Status					
Never	89.4 (87.3–91.3)	5.3 (4.0–7.0)	1.8 (1.0–2.8)	1.8 (1.0–2.8)	1.7 (1.0–2.7)
Ex-smoker	88.8 (84.7–92.1)	7.4 (4.7–10.9)	0.6 (0.0.–2.0)	1.3 (0.0–3.2)	1.9 (0.7–4.1)
Current moker	59.2 (54.7–63.6)	16.2 (13.1–19.8)	6.9 (4.8–9.5)	8.9 (6.6–11.8)	8.7 (6.4–11.6)

^1^ One or both parents born in a country other than Germany.

**Table 2 ijerph-19-04051-t002:** Descriptive sample characteristics of SHS exposure in indoor settings.

	Weighted Prevalence of SHS in Indoor Settings (95% Confidence Interval)				
Days (d) Exposed	0 d	1|2 d	3|4 d	5|6 d	all 7 d
Total					
n = 1867	75.4 (73.4–77.4)	11.4 (10.0–12.9)	5.0 (4.1–6.1)	2.3 (1.7–3.1)	5.9 (4.9–7.1)
Sex					
Male	73.5 (70.6–76.3)	12.7 (10.6–15.0)	5.1 (3.8–6.7)	3.3 (2.3–4.7)	5.4 (4.1–7.1)
Female	76.8 (74.0–79.4)	10.0 (8.2–12.1)	4.8 (3.6–6.4)	1.8 (1.1–2.9)	6.6 (5.1–8.3)
Age					
14 to 17	68.7 (56.2–79.4)	13.4 (6.3–24.0)	10.4 (4.3–20.3)	0.0 (0.0–5.4)	7.5 (2.5–16.6)
18 to 24	57.2 (49.8–64.4)	23.5 (17.6–30.3)	11.2 (7.1–16.7)	4.3 (1.9–8.3)	3.7 (1.5–7.6)
25 to 39	67.1 (62.3–71.6)	14.1 (10.9–17.9)	6.3 (4.2–9.2)	2.4 (1.2–4.4)	10.0 (7.3–13.3)
40 to 64	76.0 (72.8–79.0)	9.3 (7.4–11.6)	4.4 (3.1–6.1)	3.8 (2.5–5.4)	6.5 (4.9–8.5)
65+ years	89.1 (85.9–91.7)	6.9 (4.8–9.6)	1.3 (0.5–2.7)	0.2 (0.0–1.2)	2.5 (1.3–4.4)
Migration background					
None	76.3 (74.1–78.5)	11.0 (9.5–12.7)	4.7 (3.7–5.9)	1.7 (1.2–2.6)	6.2 (5.0–7.5)
Yes ^1^	75.1 (70.2–79.7)	9.5 (6.6–13.1)	6.8 (4.3–10.0)	3.0 (1.4–5.4)	5.6 (3.4–8.6)
Education					
High	81.1 (77.9–84.0)	11.0 (8.7–13.6)	3.7 (2.4–5.4)	1.6 (0.8–2.9)	2.5 (1.5–4.0)
Middle	74.3 (70.7–77.6)	11.0 (8.7–11.4)	5.2 (3.6–7.2)	3.4 (2.2–5.2)	6.1 (4.4–8.3)
Low	70.3 (66.2–74.3)	11.2 (8.6–14.3)	5.2 (3.5–7.5)	2.9 (1.6–4.7)	10.3 (7.8–13.2)
Income					
High	77.8 (73.7–81.5)	11.9 (9.1–15.2)	3.9 (2.3–6.1)	3.0 (1.7–5.0)	3.5 (2.0–5.6)
Middle	76.3 (73.8–78.6)	10.3 (8.7–12.2)	5.4 (4.2–6.9)	2.6 (1.8–3.7)	5.4 (4.2–6.8)
Low	63.6 (56.8–70.0)	15.7 (11.1–21.2)	4.1 (1.9–7.7)	1.4 (0.3–4.0)	15.2 (10.7–20.7)
Smoking Status					
Never	84.8 (82.6–86.9)	9.2 (7.6–11.1)	2.9 (2.0–4.1)	1.6 (1.0–2.6)	1.4 (0.8–2.3)
Ex-smoker	85.5 (80.9–89.3)	11.1 (7.8–15.3)	2.0 (0.7–4.4)	0.3 (0.0–1.9)	1.0 (0.2–2.9)
Current smoker	47.1 (42.6–51.6)	16.2 (13.1–19.8)	11.4 (8.8–14.5)	6.0 (4.1–8.5)	19.2 (15.9–23.0)

^1^ One or both parents born in a country other than Germany.

**Table 3 ijerph-19-04051-t003:** Descriptive sample characteristics of SHS exposure in outdoor settings.

	Weighted Prevalence of SHS at Outdoor Places (95% Confidence Interval)				
Days (d) Exposed	0 d	1|2 d	3|4 d	5|6 d	all 7 d
Total					
n = 1867	57.3 (55.0–59.6)	21.5 (19.2–23.0)	10.1 (8.8–11.6)	4.1 (3.3–5.1)	7.0 (5.9–8.2)
Sex					
Male	53.9 (50.7–57.2)	22.5 (19.9–25.4)	11.9 (9.9–14.1)	4.6 (3.4–6.2)	7.0 (5.5–8.9)
Female	60.2 (57.0–63.3)	20.3 (17.8–23.0)	8.1 (6.5–10.0)	4.4 (3.2–5.9)	7.1 (5.5–8.9)
Age					
14 to 17	53.8 (41.0–66.3)	24.6 (14.8–36.9)	12.3 (5.5–22.8)	4.6 (1.0–12.9)	4.6 (1.0–12.9)
18 to 24	33.9 (27.1–41.2)	34.4 (27.6–41.7)	17.2 (12.1–23.4)	8.6 (5.0–13.6)	5.9 (3.0–10.3)
25 to 39	46.6 (41.7–51.5)	21.5 (17.6–25.8)	15.1 (12.4–19.8)	5.1 (3.2–7.7)	11.7 (8.8–15.2)
40 to 64	54.7 (51.1–58.3)	22.3 (19.4–25.4)	9.4 (7.4–11.7)	5.3 (3.8–7.1)	8.4 (6.6–10.6)
65+ years	80.0 (76.0–83.5)	14.5 (11.4–18.0)	3.2 (1.8–5.2)	0.9 (0.2–2.2)	1.5 (0.6–3.1)
Migration background					
None	58.7 (56.1–61.2)	21.4 (19.4–23.7)	8.9 (7.5–10.5)	3.9 (3.0–5.0)	7,0 (5.8–8.5)
Yes ^1^	54.2 (48.7–59.7)	20.8 (16.5–25.6)	13.6 (10.1–17.7)	4.8 (2.8–7.7)	6.6 (4.2–9.9)
Education					
High	59.0 (55.2–62.7)	21.7 (18.6–25.0)	9.3 (7.2–11.7)	3.7 (2.4–5.4)	6.3 (4.6–8.4)
Middle	55.5 (51.5–59.5)	22.0 (18.8–25.5)	11.2 (8.8–13.9)	4.4 (2.9–6.3)	7.0 (5.1–9.3)
Low	58.3 (53.9–62.6)	19.0 (15.7–22.7)	9.0 (6.7–11.8)	5.1 (3.4–7.4)	8.6 (6.3–11.4)
Income					
High	57.3 (52.7–61.9)	21.4 (17.8–25.5)	9.2 (6.7–12.2)	4.2 (2.5–6.4)	7.9 (5.6–10.7)
Middle	57.1 (54.3–60.0)	22.0 (19.7–24.5)	10.5 (8.8–12.4)	4.6 (3.5–6.0)	5.7 (4.5–7.2)
Low	56.4 (43.5–56.1)	17.8 (11.4–20.7)	8.9 (4.9–11.9)	4.0 (1.6–6.6)	12.9 (7.8–15.9)
Smoking Status					
Never	68.1 (65.3–70.9)	21.4 (19.0–23.9)	5.6 (4.4–7.2)	2.8 (1.9–4.0)	2.0 (1.3–3.0)
Ex-smoker	63.8 (58.0–69.3)	22.5 (17.9–27.8)	8.9 (5.9–12.7)	3.1 (1.4–5.8)	1.7 (0.6–3.9)
Current smoker	27.9 (24.0–32.1)	20.9 (17.4–24.8)	20.3 (16.8–24.2)	9.2 (6.8–12.2)	21.6 (18.0–25.5)

^1^ One or both parents born in a country other than Germany.

**Table 4 ijerph-19-04051-t004:** Multivariable linear regression on associations between socio-epidemiological factors, smoking status, and number of days of SHS exposure in vehicles, indoors, and outdoors during the last week.

	b (95% Confidence Interval)		
	SHS Vehicle	SHS Indoors	SHS Outdoors
Total (N)	N = 1692 ^1^	N = 1703 ^1^	N = 1672 ^1^
Sex			
Male	REF	REF	REF
Female	−0.146 (−0.240 to −0.052) **	0.033 (−0.059 to 0.125)	−0.018 (−0.118 to 0.082)
Age 2			
14 to 99	−0.007 (−0.010 to −0.005) ***	−0.005 (−0.007 to −0.002) ***	−0.011 (−0.014 to −0.008) ***
Migration background			
None	REF	REF	REF
Yes ^2^	0.082 (−0.044 to 0.208)	0.021 (−0.107 to 0.149)	0.000 (−0.139 to 0.139)
Education			
High	REF	REF	REF
Middle	0.199 (0.084 to 0.314) **	0.086 (−0.025 to 0.196)	0.030 (−0.090 to 0.150)
Low	0.223 (0.091 to 0.355) **	0.309 (0.180 to 0.438) **	0.107 (−0.033 to 0.247)
Income ^3^			
EUR 0 to EUR 7000 or more	0.063 (0.002 to 0.124) *	0.000 (−0.061 to 0.061)	−0.008 (−0.074 to 0.058)
Smoking Status			
Never	REF	REF	REF
Ex-smoker	0.054 (−0.072 to 0.181)	0.018 (−0.112 to 0.148)	0.174 (0.033 to 0.316) *
Current smoker	0.665 (0.555 to 0.776) ***	1.292 (1.183 to 1.401) ***	1.216 (1.096 to 1.335) ***

^1^ Complete case analysis based on unweighted data. ^2^ One or both parents born in a country other than Germany. ^3^ Age and income were treated as continuous variables in all regression models. REF = reference category. *** *p* < 0.001, ** *p* < 0.01, * *p* < 0.05.

## Data Availability

The data underlying this study are available to researchers on reasonable request from the corresponding author (daniel.kotz@med.uni-duesseldorf.de). All proposals requesting data access will need to specify how the data is planned to be used, and all proposals will need approval of the study investigator team before data release.

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
