# Peer review of "Exposure to Tobacco Smoking in Vehicles, Indoor, and Outdoor Settings in Germany: Prevalence and Associated Factors"

_ijerph, 2022, doi:10.3390/ijerph19074051_

Round 1
Reviewer 1 Report
In this manuscript, the authors estimated the current national-level prevalence and associated
socio-epidemiological indicators of SHS exposure in vehicles, indoor, and outdoor settings in the
German population, using current data from a representative household survey. They used cross-sectional data from two waves of the DEBRA survey, conducted during the beginning of 2020. The report shows the SHS exposure percentages from vehicles, indoor settings, and outdoor settings. The authors also compared the exposure amount of different ages, education levels in Germany.
Major reviews:
In this study, the most critical factor that can determine if the report is validated or not is the sampling methodology. The authors collected specific time of a year: January to March in 2020. There could be multiple factors that are sensitive to this specific time: winter season, pandemic lockdown time, holidays…The authors need to give more explanations as to why they chose this specific time, and how representative this time is to the other months or years. Otherwise the whole conclusion would be heavily biased.
Author Response
Dear Editors and Reviewers:
We would like to express our sincere gratitude for receiving such excellent comments regarding our manuscript and for considering a revised version for publication in the IJERPH.
We hope that we have satisfactorily addressed all comments and suggestions of the reviewers.
Sincerely,
The First Author, on behalf of the co-authors of the paper
Major reviews:
In this study, the most critical factor that can determine if the report is validated or not is the sampling methodology. The authors collected specific time of a year: January to March in 2020. There could be multiple factors that are sensitive to this specific time: winter season, pandemic lockdown time, holidays…The authors need to give more explanations as to why they chose this specific time, and how representative this time is to the other months or years. Otherwise the whole conclusion would be heavily biased.
AUTHOR RESPONSE: Thank you for your comments. The fieldwork for the current study was conducted in January 2020 and February/March 2020. Unfortunately, it was not feasible to collect data during all other months of the year. Were therefore do not know whether the level of second-hand smoke (SHS) exposure varies between different time points of the year, and neither do we have data regarding the “average” SHS exposure of the year. We do not think this biased our findings because there is no reason to assume that the association between SHS exposure and socio-epidemiological indicators is dependent on the month of the year. However, we cannot rule out that the absolute level of SHS exposure is different in different months of the year, which limits the generalizability of our findings. We agree this is an important discussion point which we should add to the Limitations part as follows.
CHANGES TO THE MANUSCRIPT (lines 315 ff): Discussion, Limitations and strengths: “Our fieldwork was conducted in the period between January and March 2020 and did not cover the entire year. The level of SHS exposure may be different in different months of a year, e.g., due to the weather or holidays. We do not expect this to have an influence on the associations between SHS exposure and the socio-epidemiological under study. However, our estimated prevalence rates of SHS exposure may be different from the true annual average.”
Reviewer 2 Report
Review of “Exposure to tobacco smoking in vehicles, indoor, and outdoor settings in Germany: prevalence and associated factors”.
This paper intends to estimate the current national-level prevalence and associated socio-epidemiological indicators of SHS exposure in vehicles, indoor, and outdoor settings in German. The topic is of great importance and broad interest for the readers. I have some questions for consideration.
- The paper includes “vehicles” inside the “indoor” (such as Line 11), so does the title need to be modified to avoid misunderstanding of readers?
- A person will stay in different settings, i.e., vehicles, indoor and outdoor. How do the authors separate them? This is not well understood in the paper.
- In the results part of the paper, there are too few descriptions and explanations of the results. For example, the analysis and discussion of the results in Table 1-4 should be appropriately supplemented.
- There are some formatting problems in the writing of the paper, such as:
(1) Line 17: One more “.”.
(2) The citation format of references is not standard, which affects reading (e.g., Line 31).
Author Response
Dear Editors and Reviewers:
We would like to express our sincere gratitude for receiving such excellent comments regarding our manuscript and for considering a revised version for publication in the IJERPH.
We hope that we have satisfactorily addressed all comments and suggestions of the reviewers.
Sincerely,
The First Author, on behalf of the co-authors of the paper
The paper includes “vehicles” inside the “indoor” (such as Line 11), so does the title need to be modified to avoid misunderstanding of readers?
AUTHOR RESPONSE: Thank you for your helpful comments. Here it is rather a matter of phrasing and shortening the abstract. Our statistical analyses and results part differentiates the three settings mentioned in the title of the paper: vehicles, indoor settings, and outdoor settings. We have revised the abstract accordingly.
CHANGES TO THE MANUSCRIPT (line 11): Abstract: “Little is known on whether secondhand smoke (SHS) exposure in vehicles, indoor, and outdoor settings are similarly patterned in terms of different socio-epidemiological indicators in Germany.”
A person will stay in different settings, i.e., vehicles, indoor and outdoor. How do the authors separate them? This is not well understood in the paper.
AUTHOR RESPONSE: We agree. It is possible that the same person had been exposed to SHS in different settings. However, the aim of our analysis was not to measure the total level or burden of SHS exposure from all settings together. We asked respondents for all three settings on how many of the past 7 days they had been in vehicles, indoor, and outdoor settings and inhaled the smoke of someone who smoked tobacco, i.e., cigarette or cigar or pipe. Our aim was to estimate the likelihood of SHS exposure in each of the different settings. We have revised our paper to clarify this aspect.
CHANGES TO THE MANUSCRIPT (lines 153 ff): Methods/Outcome Measures: “It is possible that the same respondent reported SHS exposure in more than one setting. Hence, our outcomes measured the likelihood of SHS exposure in each of the different settings, rather than the total level or burden of SHS exposure from all settings together.”
In the results part of the paper, there are too few descriptions and explanations of the results. For example, the analysis and discussion of the results in Table 1-4 should be appropriately supplemented.
AUTHOR RESPONSE: Thank you. We suggest to add a few additional explanations and descriptions in the Results section. A deeper interpretation of the results is dedicated to the Discussion section.
CHANGES TO THE MANUSCRIPT (lines 214 ff): Results: “People in the age group of 25 to 39 years were most frequently exposed to SHS at all seven days of the past week in vehicles (7%), indoor (10%) and outdoor (12%) settings. In current smokers, exposure rates at all seven days in the three settings were 9%, 19% and 22%, respectively.”
There are some formatting problems in the writing of the paper, such as:
(1) Line 17: One more “.”.
(2) The citation format of references is not standard, which affects reading (e.g., Line 31).
AUTHOR RESPONSE: Thank you for spotting these errors. We have modified these accordingly.
Reviewer 3 Report
General comments:
The manuscript presents the results of a cross-sectional study estimating the current national-level prevalence and associated socio-epidemiological indicators of SHS exposure in vehicles, indoor, and outdoor settings in the German population. Although authors disclose some limitations in the study, the objectives are clear and well defined, the methods used are appropriate, and the conclusions drawn are suitably supported by the results. The manuscript highlights the role of sociodemographic and socioeconomic characteristics on SHS exposure.
Specific comment:
- Authors should consider using sociodemographic and socioeconomic characteristics rather than socio-epidemiological indicators
- Authors should also discuss the role of COVID-19 (stress associated to COVID-19 and lockdown) on current smokers as well as on SHS. COVID-19 may change the smoking behaviors, and therefore the representativeness of the results.
- Line 75, “one”, please check the sentence
- Why did the authors exclude e-cigarettes from the manuscript? e-cigarettes are also important sources of air pollutants and exposure to e-cigarettes has also been associated with adverse health effects
- Why did the authors use different statistical software to perform the statistical analyses?
- Authors should consider deleting the sentence from lines 199 to 201, this sentence reports the same results as those presented in the following lines (201-203) and these results are clearer considering the aim of the study
- Check the results in line 203
- In discussion, it will be interesting and relevant to report some individual-level interventions to avoid or decrease the exposure to SHS
Author Response
Dear Editors and Reviewers:
We would like to express our sincere gratitude for receiving such excellent comments regarding our manuscript and for considering a revised version for publication in the IJERPH.
We hope that we have satisfactorily addressed all comments and suggestions of the reviewers.
Sincerely,
The First Author, on behalf of the co-authors of the paper
Authors should consider using sociodemographic and socioeconomic characteristics rather than socio-epidemiological indicators
AUTHOR RESPONSE: Thank you for your comments. We agree that this may be appropriate as well. However, social epidemiology includes sociodemographic and socioeconomic characteristics as main pillars, which is why we believe that our terminology is appropriate for the paper (and it is also a bit more concise). We explain in the Methods section the exact socio-epidemiological indicators we are using in our analyses.
Authors should also discuss the role of COVID-19 (stress associated to COVID-19 and lockdown) on current smokers as well as on SHS. COVID-19 may change the smoking behaviors, and therefore the representativeness of the results.
AUTHOR RESPONSE: This is a very interesting thought. However, the data for the current study were conducted prior to the start of the COVID-19 pandemic in Germany.
Line 75, “one”, please check the sentence
AUTHOR RESPONSE: We have checked the sentence and think the wording is grammatically correct.
Why did the authors exclude e-cigarettes from the manuscript? e-cigarettes are also important sources of air pollutants and exposure to e-cigarettes has also been associated with adverse health effects
AUTHOR RESPONSE: We believe that secondhand exposure to tobacco smoke should not be mixed with exposure to e-cigarettes or other sources of air pollution. The focus of the paper respectively exposure was explicitly on exposure to tobacco smoke, i.e., cigarette or cigar or pipe.
Why did the authors use different statistical software to perform the statistical analyses?
AUTHOR RESPONSE: SPSS is limited in calculating confidence intervals for prevalence respectively relative frequencies. That is why we used R Studio for this purpose.
Authors should consider deleting the sentence from lines 199 to 201, this sentence reports the same results as those presented in the following lines (201-203) and these results are clearer considering the aim of the study
Check the results in line 203
AUTHOR RESPONSE: This is good suggestion, thank you! We have rephrased both sentences as follows.
CHANGES TO THE MANUSCRIPT (lines 204f): Results: “Tables 1 to 3 display that the reported prevalence of SHS exposure during the last seven days was 19% in vehicles, 25% in indoor settings, and 43% in outdoor settings.”
In discussion, it will be interesting and relevant to report some individual-level interventions to avoid or decrease the exposure to SHS
AUTHOR RESPONSE: We argue and conclude in the paper that population-level interventions are more important to avoid or decrease the exposure to SHS. Progressive implementation in Germany of the measures drawn up in the WHO Framework Convention on Tobacco Control such as smoke-free car legislation, advertising bans, comprehensive smoke-free policies in the hospitality sector, and increased tax policies should be given priority in terms of health policy.
Round 2
Reviewer 1 Report
The comments have been properly addressed. I encourage to publish this manuscript.